



# Long-Term Global Ground Heat Flux and Continental Heat Storage from Geothermal Data

Francisco José Cuesta-Valero[1,2], Almudena García-García[1,2], Hugo Beltrami[1,3], J. Fidel González-Rouco[4], and Elena García-Bustamante[5]

[1]Climate & Atmospheric Sciences Institute, St. Francis Xavier University, Antigonish, NS, Canada.
[2]Environmental Sciences Program, Memorial University of Newfoundland, St. John's, NL, Canada.
[3]Department of Earth Sciences, St. Francis Xavier University, Antigonish, Nova Scotia, Canada.
[4]Universidad Complutense de Madrid, 28040 Madrid, Spain.
[5]Centro de Investigaciones Energéticas, Medioambientales y Tecnológicas (CIEMAT), 28040 Madrid, Spain.

**Correspondence:** Hugo Beltrami (hugo@stfx.ca)

**Abstract.** Energy exchanges among climate subsystems are of critical importance to determine the climate sensitivity of the Earth's system to greenhouse gases, to quantify the magnitude and evolution of the Earth's energy imbalance, and to project the evolution of future climate. Thus, ascertaining the magnitude and change of the Earth's energy partition within climate subsystems has become urgent in recent years. Here, we provide new global estimates of changes in ground surface

temperature, ground surface heat flux and continental heat storage derived from geothermal data using an expanded database and new techniques. Results reveal markedly higher changes in ground heat flux and heat storage within the continental subsurface than previously reported, with land temperature changes of 1 K and continental heat gains of around 12 ZJ during the last part of the 20[th] century relative to preindustrial times. Half of the heat gain by the continental subsurface since 1960 have occurred in the last twenty years.

## 1  Introduction

Climate change is consequence of the current radiative imbalance at top-of-the-atmosphere, which delivers an excess amount of energy to the Earth's system in comparison with preindustrial conditions (Hansen et al., 2011; Stephens et al., 2012; Lembo et al., 2019). Nonetheless, the energy imbalance presents and interhemispheric asymmetry, being larger in the southern hemisphere (Loeb et al., 2016; Irving et al., 2019). This asymmetry causes an increase in the heat uptake by the ocean surface

in the southern hemisphere in comparison with the ocean heat uptake in the northern hemisphere. Hence, a cross-equatorial northward transport of heat emerges to compensate this asymmetry (Lembo et al., 2019), in addition to the the global meridional heat transport caused by the different radiation levels reaching the tropical and polar oceans (Trenberth et al., 2019). The hemispheric distribution of heat uptake, heat storage and heat transport is expected to change under different emission scenarios (Irving et al., 2019), meaning that characterizing where the heat enters the system (uptake), where the heat is allocated

(storage), and where the heat is redistributed (transport), is of critical importance to understand the evolution of climate change.





The vast majority of excess heat due to the Earth's energy imbalance is stored in the ocean (84-93%), followed by the cryosphere (4-7%) and the continental subsurface (2-5%), with the atmosphere showing the smaller heat storage term (1-4%) (Levitus et al., 2005; Church et al., 2011). Therefore, extensive resources are devoted to monitor and understand the evolution of the ocean heat content, since it is also an indirect method to study the magnitude and variations of the energy
imbalance at top-of-atmosphere (Palmer et al., 2011; Palmer and McNeall, 2014; Johnson et al., 2016; Riser et al., 2016; von Schuckmann et al., 2016). The rest of the components of the climate system have relevant roles in the Earth's energy budget, despite their small contribution to storage (Levitus et al., 2005; Church et al., 2011; Hansen et al., 2011; von Schuckmann et al., 2016). For instance, some energy-dependent processes are permafrost stability and the associated permafrost carbon feedback (MacDougall et al., 2012; Hicks Pries et al., 2017), changes in circulation patterns (Tomas et al., 2016; Screen
et al., 2018) and sea level rise (Jacob et al., 2012; Vaughan et al., 2013; Dutton et al., 2015). The additional energy in the atmosphere, cryosphere and continental subsurface also affects near-surface conditions, having important consequences for society. Increases in atmospheric heat content produce warmer surface air temperature and larger amounts of water content within the atmosphere that can impact crop yields, and consequently global food security (Lloyd et al., 2011; Rosenzweig et al., 2014; Phalkey et al., 2015; Campbell et al., 2016) as well as degrading human health due to heat stress (Sherwood and
Huber, 2010; Matthews et al., 2017; Watts et al., 2019). Floods induced by extreme precipitation events, which frequency and intensity are affected by the amount of water in the atmosphere, as well as floods induced by sea level rise caused by the thermal expansion of the ocean and melting of Greenland and Antarctica ice sheets, are likely to impact human settlements (McGranahan et al., 2007; Kundzewicz et al., 2014). Furthermore, all these alterations of surface environmental conditions may enhance the spread of diseases (Levy et al., 2016; McPherson et al., 2017; Wu et al., 2016; Watts et al., 2019), among
other potential risks.

Long-term global estimates of heat storage within the continental subsurface (ground heat content, GHC) have been previously estimated from borehole temperature profile (BTP) measurements. Changes in the energy balance at the land surface add or remove heat from the upper continental crust, changing the long-term subsurface equilibrium temperature profile (Beltrami, 2002b). Such temperature changes propagate through the ground by conduction, and are recorded in the subsurface as
perturbations on the quasi-steady state vertical temperature profile. Borehole climatology consists in estimating variations in ground surface temperature and heat flux from these recorded alterations in the subsurface thermal regime. Ground Surface Temperature Histories (GSTHs) and Ground Heat Flux Histories (GHFHs) have been retrieved from BTP measurements both at regional and at hemispheric scales for multi-century to multi-millennial time periods (Lane, 1923; Cermak, 1971; Beck, 1977; Vasseur et al., 1983; Lachenbruch and Marshall, 1986; Huang et al., 2000; Harris and Chapman, 2001; Roy et al., 2002;
Beltrami and Bourlon, 2004; Hartmann and Rath, 2005; Beltrami et al., 2006; Hopcroft et al., 2007; Chouinard and Mareschal, 2009; Davis et al., 2010; Barkaoui et al., 2013; Demezhko and Gornostaeva, 2015; Jaume-Santero et al., 2016; Pickler et al., 2016), constituting an useful reference for evaluating climate simulations performed by Coupled General Circulation Models (CGCMs) beyond the observational period (González-Rouco et al., 2009; Stevens et al., 2008; MacDougall et al., 2010; Cuesta-Valero et al., 2016; García-García et al., 2016; Cuesta-Valero et al., 2019), as well as for evaluating reconstructions





derived from other paleoclimate data (Fernández-Donado et al., 2013; Masson-Delmotte et al., 2013; Jaume-Santero et al.,
2016; Beltrami et al., 2017).

Previous global estimates of GHC, GHFHs and GSTHs have been retrieved from BTP measurements nearly two decades
ago (Pollack et al., 1998; Huang et al., 2000; Beltrami et al., 2002; Beltrami, 2002a; Pollack and Smerdon, 2004), including a
limited characterization of uncertainties. Meanwhile, advances in borehole methodology have allowed to assess the uncertainty

in borehole reconstructions induced by a series of factors: the presence of advection and freezing phenomena, the sampling rate
and the depth range used in the determination of the quasi-equilibrium profile, the depth of the log, the different logging dates
of the profiles, the noise in the measured profile, the model resolution for obtaining stable solutions, the spatial distribution of
borehole measurements, and the transient variations in the subsurface thermal regime due to the end of the last glacial cycle
(Bodri and Cermak, 2005; Hartmann and Rath, 2005; Reiter, 2005; González-Rouco et al., 2006; Mottaghy and Rath, 2006;

González-Rouco et al., 2009; Rath et al., 2012; Beltrami et al., 2015a, b; García-García et al., 2016; Jaume-Santero et al., 2016;
Beltrami et al., 2017; Melo-Aguilar et al., 2019). These advances together with the availability of new BTP measurements make
necessary an update of the global long-term evolution of ground heat content from borehole data.

Here, we use an expanded borehole database to estimate global GSTHs, GHFHs, and GHC within the continental subsurface
for the last four centuries. Surface temperature and heat flux histories are retrieved from each BTP using a Singular Value

Decomposition (SVD) algorithm, one of the standard borehole methodologies employed in previous analyses (Beltrami et al.,
2002; Beltrami, 2002a), as well as a new approach based on generating an ensemble of inversions for each temperature profile
to explore additional sources of uncertainty unaddressed in previous global borehole reconstructions.

We find higher values of surface temperature, ground heat flux at the surface and ground heat content from borehole data
than previously reported. These results imply that a larger amount of the additional energy gained by the Earth system is

allocated within the continental subsurface than previously thought, reinforcing the necessity of monitoring the continental
heat storage and the need for improving the representation of the land component of the energy budget within long-term
climate simulations. These results also support previous estimates of temperature change since preindustrial times based on
meteorological observations and CGCM simulations, using estimates from an independent source of data and considering the
most distant period of time to determine preindustrial conditions to our knowledge.

## 2  Theory

### 2.1  Subsurface Temperature Profile

In borehole climatology, the continental subsurface is typically represented as a semi-infinite homogenous half-space without
internal sources of heat, where energy exchanges at the land surface and heat flux from the Earth's interior are considered as the
upper and bottom boundary conditions. The local subsurface thermal regime is, therefore, the result of a balance between the

surface thermal state and the thermal conditions of the Earth's interior. If surface conditions remain stable at long time scales,
the subsurface thermal regime would be at a quasi-equilibrium since the flux from the Earth's interior is constant at geological
time scales (million years). Thereby, the subsurface temperature profile can be expressed as the superposition of the transient



temperature due to changes in the surface conditions ($T_t$) relative to the long-term quasi-equilibrium state (Carslaw and Jaeger, 1959):

$$T(z) = T_0 + q_0 R(z) + T_t(z), \tag{1}$$

where $z$ is depth, $T_0$ is the long-term surface temperature, $q_0$ is the heat flux from the Earth's interior, and $R(z) = \int_0^z \frac{\mathrm{d}z'}{\lambda(z')}$ is the thermal depth, which depends on the thermal conductivity ($\lambda$) of the ground (Bullard and Schonland, 1939). Since measurements of thermal conductivity profiles are scarce and the measured profiles typically display variations around a constant value with depth, the thermal conductivity can be assumed to be constant and Equation 1 can be rewritten as

$$T(z) = T_0 + \Gamma \cdot z + T_t(z), \tag{2}$$

with $\Gamma = \frac{q_0}{\lambda}$ the equilibrium subsurface thermal gradient. The term $T_0 + \Gamma \cdot z$ in Equation 2 describes the quasi-equilibrium temperature profile, and can be determined from the deepest part of a BTP - that is, the least affected part of the log by recent perturbations of the energy balance at the surface.

The propagation of temperature variations in a one-dimensional, homogenous, isotropic medium without internal sources of heat is governed by the heat diffusion equation

$$\frac{\partial T}{\partial t} = \kappa \frac{\partial^2 T}{\partial z^2}, \tag{3}$$

where $T$ is temperature, $t$ is time, $\kappa$ is the thermal diffusivity of the medium and $z$ is the spatial dimension. An instantaneous change in surface temperature ($\Delta T_0$) is propagated through the ground as described in Equation 3, altering the quasi-equilibrium temperature profile with time following (Carslaw and Jaeger, 1959)

$$T(z,t) = \Delta T_0 \cdot \mathrm{erfc}\left(\frac{z}{2\sqrt{\kappa t}}\right), \tag{4}$$

where $\mathrm{erfc}$ is the complementary error function, and $t$ is time since the surface temperature change. A series of surface temperature perturbations will propagate through the ground as the superposition of transient variations of the long-term subsurface thermal regime:

$$T_t(z) = \sum_{i=1}^{N} \Delta T_i \left[\mathrm{erfc}\left(\frac{z}{2\sqrt{\kappa t_i}}\right) - \mathrm{erfc}\left(\frac{z}{2\sqrt{\kappa t_{i-1}}}\right)\right], \tag{5}$$

where $\Delta T_i$ are changes in surface temperature at $i$ time step. Equation 5 is also the solution of the forward problem: given an upper (surface) boundary condition, this equation describes the perturbation of the subsurface temperature profile in response to a temporal series of ground surface temperature changes (Lesperance et al., 2010).

## 2.2 Subsurface Flux Profile

Since the conductive heat flux ($q$) in an isotropic medium is related to the temperature gradient of the subsurface temperature profile by Fourier's equation

$$q = -\lambda \frac{\partial T}{\partial z}, \tag{6}$$





the propagation of heat flux through a one-dimensional, homogenous medium without internal sources of heat satisfies:

$$\frac{\partial q}{\partial t} = \kappa \frac{\partial^2 q}{\partial z^2}. \tag{7}$$

That is, the propagation of both temperature and heat flux through the ground is governed by the diffusion equation (Carslaw and Jaeger, 1959; Turcotte and Schubert, 2002). As in the case of temperature profiles, the heat flux profile can be expressed as

$$q(z) = q_0 + q_t(z), \tag{8}$$

where $q_0$ is the equilibrium geothermal flux from the Earth's interior. Therefore, alterations in the subsurface equilibrium flux profile due to an instantaneous perturbation of the long-term surface flux ($\Delta q_0$) can be expressed as

$$q(z,t) = \Delta q_0 \cdot \text{erfc}\left(\frac{z}{2\sqrt{\kappa t}}\right), \tag{9}$$

where $t$ is time since the perturbation. A series of perturbations of the surface flux generates a superposition of transient variations of the long-term subsurface thermal gradient as

$$q_t(z) = \sum_{i=1}^{N} \Delta q_i \left[\text{erfc}\left(\frac{z}{2\sqrt{\kappa t_i}}\right) - \text{erfc}\left(\frac{z}{2\sqrt{\kappa t_{i-1}}}\right)\right], \tag{10}$$

mirroring the forward model for surface temperature variations described in Equation 5 and representing the solution of the forward problem for variations in surface heat flux (Beltrami, 2001; Beltrami et al., 2006).

## 2.3 Inversion Problem

The inversion problem consists in retrieving the past ground surface temperature histories that generated the observed temperature perturbation profiles, or the ground heat flux histories that generated the heat flux anomaly profiles. A system of equations can be derived by combining Equations 2 and 5 for the temperature case, and Equations 8 and 10 for the heat flux case, with the solution of such systems yielding an estimate of the past long-term evolution of surface temperature and surface heat flux, respectively (Vasseur et al., 1983; Beltrami et al., 1992; Mareschal and Beltrami, 1992; Shen et al., 1992; Beltrami, 2001; Hartmann and Rath, 2005). This system can be expressed for the temperature case as:

$$\begin{pmatrix} T_t(z_1) \\ \vdots \\ T_t(z_i) \\ \vdots \\ T_t(z_{N_z}) \end{pmatrix} = \begin{pmatrix} M_{1,1} & \cdots & M_{1,j} & \cdots & M_{1,N_t} \\ \vdots & \ddots & \vdots & \ddots & \vdots \\ M_{i,1} & \cdots & M_{i,j} & \cdots & M_{i,N_t} \\ \vdots & \ddots & \vdots & \ddots & \vdots \\ M_{N_z,1} & \cdots & M_{N_z,j} & \cdots & M_{N_z,N_t} \end{pmatrix} \begin{pmatrix} \Delta T_1 \\ \vdots \\ \Delta T_j \\ \vdots \\ \Delta T_{N_t} \end{pmatrix} \tag{11}$$

were $T_t(z_i)$ are the temperature anomalies at the depth $z_i$, and $\Delta T_i$ are the step change in surface temperature to be determined, that is the proposed inversion model. The elements $M_{i,j}$ are defined from the forward model (Equation 5)

$$M_{i,j} = \text{erfc}\left(\frac{z_i}{2\sqrt{\kappa t_j}}\right) - \text{erfc}\left(\frac{z_i}{2\sqrt{\kappa t_{j-1}}}\right). \tag{12}$$



Note that a similar system can be written in terms of heat flux using Equation 10. The rank of the system is given by the number of time steps in the proposed inversion model ($N_t$), and is generally smaller than the number of measurements in the profile ($N_z$). That is, there are more equations than parameters in the system, thus both the temperature and heat flux systems
are overdetermined. Therefore, these systems are solved using a Singular Value Decomposition algorithm (Lanczos, 1961) as the one described in Mareschal and Beltrami (1992) and Clauser and Mareschal (1995).

The system in Equation 11 can be expressed as a matrix equation of the form:

$$\mathbf{T}_{obs} = \mathbf{M}\mathbf{T}_{model}, \tag{13}$$

where $\mathbf{T}_{obs}$ is the data vector (anomaly temperature profile of heat flux profile), $\mathbf{M}$ is the matrix containing the coefficients
given by Equation 12 (or the equivalent expression for the case of heat flux), and $\mathbf{T}_{model}$ is a vector containing the step change model to be determined. The SVD algorithm decomposes the matrix of coefficients as

$$\mathbf{M} = \mathbf{U}\mathbf{S}\mathbf{V}^T, \tag{14}$$

with $\mathbf{U}$ and $\mathbf{V}$ orthonormal matrices of dimension $N_z \times N_z$ and $N_t \times N_t$, respectively, and $\mathbf{S}$ a rectangular matrix ($N_z \times N_t$) containing the eigenvalues $\alpha_j$ in the diagonal. Therefore, the general solution can be expressed as:

$$\mathbf{T}_{model} = \mathbf{M}^{-1}\mathbf{T}_{obs} = \mathbf{V}\mathbf{S}^{-1}\mathbf{U}^T\mathbf{T}_{obs}. \tag{15}$$

However, the solution of Equation 15 is dominated by noise from small eigenvalues, as the only non-zero elements of $\mathbf{S}^{-1}$ are the inverse of the eigenvalues in the diagonal of the matrix (Mareschal and Beltrami, 1992). Accordingly, small eigenvalues need to be removed from $\mathbf{S}^{-1}$ (i.e., are replaced by zeros) for stabilizing the solution, but at the cost of losing temporal resolution in the model.

## 3 Analysis

### 3.1 Borehole Data

Borehole Temperature Profiles (BTPs) were collected from four databases. The National Oceanic and Atmospheric Administration (NOAA) server (NOAA, 2019) contains global data; the database presented in Jaume-Santero et al. (2016) includes data for North America; logs from Tasmania were retrieved from Suman et al. (2017); and measurements from Chile were obtained
from Pickler et al. (2018). Profiles from all databases were screened to avoid repetitions, resulting in 1266 independent logs in total.

Nonetheless, not all these BTPs are employed in the analysis. A process for selecting suitable logs is applied, based on trimming the maximum depth of the available BTPs and requiring a certain number of measurements at critical depth ranges. Thereby, all BTPs used here must include at least one temperature measurement between 15 m and 100 m, and between 250 m
and 310 m. Profiles containing less than three measurements between 200 m and 300 m were discarded since it was impossible to perform a linear regression analysis to determine the quasi-equilibrium profile (see Section 3.3 below). All remaining logs





were truncated from 15 m to 300 m depth, resulting in 1079 logs selected for this analysis. Thereby, we ensure that the profiles include information from the logging year to several centuries back in time and cover the same time span, since the relationship between time ($t$) required for a change in the surface energy balance to reach a certain depth ($z$) can be approximated as

(Carslaw and Jaeger, 1959; Pickler et al., 2016; Cuesta-Valero et al., 2019):

$$t \approx \frac{z^2}{4\kappa}, \qquad (16)$$

This depth filtering constitutes the main methodological difference in comparison with previous borehole studies (Beltrami et al., 2002; Beltrami, 2002a, including[]), since those assessments analyzed all available logs independently of their depth range, thus mixing temporal references. However, recent works have shown that using subsurface profiles with different depths

affects the estimated GSTHs (González-Rouco et al., 2009; Beltrami et al., 2015b; Melo-Aguilar et al., 2019). This issue is avoided here by the selection criteria applied to the assembled BTP database. Additionally, BTPs were measured at different dates, but the logging year of the profiles had been taken intro account only in a small number of works (e.g., González-Rouco et al., 2009; Jaume-Santero et al., 2016; Melo-Aguilar et al., 2019). We aggregate the retrieved GSTHs and GHFHs from BTPs considering the logging date of each borehole profile (Figure 1), thus the number of borehole inversions available for analysis

varies with time.

### 3.2 Surface Air Temperature Data

Meteorological measurements of Surface Air Temperature (SAT) from the Climate Research Unit (CRU) at East Anglia university are also used in this study to compare with borehole estimates. Mean global SAT anomalies relative to 1961-1990 Common Era (CE) from the CRU TS 4.01 product (Harris et al., 2014) are employed to compare with GSTHs retrieved from

borehole profiles. Results for the entire CRU spatial and temporal domains are provided from 1901 CE to 2016 CE, as well as results considering only locations and dates containing borehole inversions.

### 3.3 Inversion of Borehole Temperature Profiles

#### 3.3.1 Standard Inversions

We invert the same truncated BTPs to obtain GSTHs considering the uncertainty from the determination of the equilibrium

profile, as a reference to compare with the uncertainty estimates of recent works using the same SVD algorithm (Beltrami et al., 2015a; Jaume-Santero et al., 2016; Pickler et al., 2016, 2018). In this case, all logs are inverted considering a model based on a thermal conductivity of $3\,\mathrm{W\,m^{-1}\,K^{-1}}$, a volumetric heat capacity of $3 \times 10^6\,\mathrm{J\,m^{-3}\,K^{-1}}$, and thus a thermal diffusion of $1 \times 10^{-6}\,\mathrm{m^2\,s^{-1}}$. The same SVD algorithm used in Beltrami (2002a) and Beltrami et al. (2002) is applied to generate the GSTHs for three step change models, since there is no preferential inversion model. All BTPs are inverted using models based

on step changes of 25, 40 and 50 years to reconstruct the surface signal for 400 years before the logging date of the profile (i.e., inversion models of 16, 10 and 8 time steps, respectively), with all inversions including the four highest eigenvalues. We regard these as the Standard inversion approach and will serve as a reference to the methods described in Section 3.3.2.





The equilibrium temperature profile is estimated in order to obtain the anomaly profile that is inverted by the SVD algorithm. The equilibrium profile is estimated from the deepest part of each truncated BTP, since that is the zone least affected by the recent climate change signal (grey zone in Figure 2a). A linear regression analysis of the lowermost $100\,\mathrm{m}$ of each profile (from $200\,\mathrm{m}$ to $300\,\mathrm{m}$ depth in our analysis, straight lines in Figure 2a) is performed to estimate the values determining the quasi-equilibrium temperature profile, that is, the long-term surface temperature ($T_0$) and the equilibrium geothermal gradient ($\Gamma$). We use the last hundred meters and not a longer depth range to reach a balance between the characterization of noise and retrieving as much climatic information as possible from each log (Beltrami et al., 2015a). The anomaly profile is then retrieved by subtracting the quasi-equilibrium temperature profile from the measured log (black dots in Figure 2b). Additionally, the uncertainties in the slope ($\Gamma$) and intercept ($T_0$) values allow to obtain two extremal temperature anomaly profiles representing the 95% confidence interval (two standard deviations) of the anomaly profile (red and blue dots in Figure 2b). The inversion of these additional anomaly profiles provide the 95% confidence interval of the retrieved GSTHs from each borehole. We do not invert the heat flux profiles using this approach, but provide surface flux estimates from the retrieved surface temperature histories to compare with Beltrami (2002a) and Beltrami et al. (2002) (see Section 3.4 for details).

### 3.3.2 Perturbed Parameter Inversions

Although the inversion approach used in previous studies was successful in retrieving the past long-term evolution of ground surface temperatures and ground heat fluxes at BTP locations, several sources of uncertainty remained unaddressed. Here, we use a new approach based on generating an ensemble of inversions using the SVD algorithm described in Mareschal and Beltrami (1992) for each borehole profile to account for as many sources of uncertainty as possible. The ensemble contains inversions retrieved by considering a range of values for the thermal properties, different number of eigenvalues in the SVD algorithm, as well as the inversions of the two additional anomaly profiles generated from the estimate of the quasi-equilibrium temperature profile. Thereby, three sources of uncertainty are considered in the analysis, expanding the methodology of previous studies based on BTP inversions performed with the same SVD algorithm (Beltrami et al., 2015a; Jaume-Santero et al., 2016; Pickler et al., 2016, 2018). Additionally, all BTPs are inverted using the three different inversion models used in the Standard approach. We name this new approach as the Perturbed Parameter Inversion (PPI thereinafter) due to the similarities with the generation of perturbed parameter ensembles in climate modeling (e.g., Collins et al., 2011).

The PPI approach considers the three anomaly profiles estimated from the uncertainty in determining the subsurface equilibrium profile as in the Standard approach (e.g., Jaume-Santero et al., 2016, and section above). Each of the these anomaly profiles is inverted using different values of thermal conductivity ($\lambda$) and volumetric heat capacity ($\rho C$). The values of thermal conductivity considered in this analysis are 2.5, 3 and $3.5\,\mathrm{W\,m^{-1}\,K^{-1}}$, while the values for volumetric heat capacity are 2.5, 3 and $3.5 \times 10^6\,\mathrm{J\,m^{-3}\,K^{-1}}$. That is, the typical values of $3\,\mathrm{W\,m^{-1}\,K^{-1}}$ and $3 \times 10^6\,\mathrm{J\,m^{-3}\,K^{-1}}$ for the conductivity and heat capacity, respectively, as well as two extremal cases to account for plausible variations of thermal properties. The combination of each pair of conductivities and heat capacities yields a series of 9 values for thermal diffusivity ranging between 0.7 and $1.4 \times 10^{-6}\,\mathrm{m^2\,s^{-1}}$. Additionally, estimates obtained for the three inversion models use different numbers of eigenvalues to retrieve the surface signal, attending to the sensitivity of the SVD algorithm to small eigenvalues and to the length of each time



step in the inversion model (Hartmann and Rath, 2005; Melo-Aguilar et al., 2019). Thus, inversions based on the 25 yr step change model use the highest 3, 4 and 5 eigenvalues, inversions based on the 40 yr step change model use the highest 2, 3 and 4 eigenvalues, and inversions based on the 50 yr step change model use the highest 2, 3 and 4 eigenvalues.

Therefore, the PPI ensemble generated from each original borehole temperature profile consists of 243 different GSTH inversions (see the case for GHFHs below). All these inversions are then propagated using a purely conductive forward model in order to obtain synthetic BTPs as described in Equation 5, which are compared with the original anomaly profiles (Figure 2c). This allows to evaluate the performance of the different parameter variants in the inversion and to attribute relative weights to them. Root mean squared errors (RMSEs) between the anomaly profiles and the synthetic profiles generated from the
inversions are computed to assign a weight to each inversion following a gaussian function as in Knutti et al. (2017):

$$w_i = \exp\left\{\frac{-\text{RMSE}_i^2}{\sigma^2}\right\}, \tag{17}$$

where $w_i$ is the weight associated to the $i$th inversion, and $\sigma$ is a parameter determining which RMSEs are deemed as large and which are deemed as small. We select the typical error in BTP measurements ($\sigma = 50$ mK) as criterium to assess how each inversion should be weighted, that is, to evaluate which RSMEs are large and which are small.

Thus, each inversion is classified according to the realism of its associated synthetic anomaly profile. Nevertheless, unrealistic solutions may arise as result of the broad range of parameters and inversion models considered even after weighting each inversion. Hence, we introduce here a new additional criterium to asses all the 243 inversions per BTP based on the variability of surface air temperature measurements as a guide. A temperature change in an inverted GSTH is considered unrealistic if it is larger than the maximum change obtained from the histogram of temperature variations between consecutive time steps from
the CRU data. This histogram is created by aggregating temperature changes between consecutive time steps after averaging the original temperature series at each grid cell in temporal windows of 25 years (i.e, running means of 25 years, Figure S1). The averaging of the original temperature series is necessary to remove high-frequency variability that is not present in GSTHs from BTP inversions. That is, a GSTH is deemed as unrealistic and removed from the analysis if the temperature change between at least one pair of consecutive time steps is larger than 2.57 K for the three inversion models. The 5th, 50th, and 95th
weighted percentiles are eventually estimated from the ensemble of remaining inversions (Figure 2d) for each borehole profile. The ensemble containing the weighted percentiles from GSTHs from all BTPs is called PPIT ensemble hereinafter.

The same approach is applied to the corresponding heat flux profiles to retrieve ground heat flux histories from borehole data. The heat flux profiles are generated from the three estimated temperature anomaly profiles for each measured log using the Fourier's equation (Equation 6) as

$$q_i = -\lambda \frac{T_{i+1} - T_i}{z_{i+1} - z_i}. \tag{18}$$

Those profiles are then inverted using the PPI approach described above. The thermal conductivity for estimating the heat flux profile is set to match the values used for each perturbed parameter inversion. Thereby, we obtain 243 heat flux histories for each original log, which are compared to the corresponding flux anomaly profile (using Equation 10) and weighted as in the case of temperature histories. Changes in GHFHs are compared to the histogram created by aggregating heat flux changes





estimated from the CRU temperature data and Equation 19 in order to discard unrealistic heat flux histories. As in the case of temperature changes, heat flux changes between consecutive time steps are aggregated after averaging the original heat flux series from each grid cell over temporal windows of 25 years (Figure S1). Surface heat flux histories are deemed unrealistic if the difference between at least one pair of consecutive time steps is larger than $0.51 \, \mathrm{W \, m^{-2}}$ for the three inversion models. The ensemble containing the 5th, 50th and 95th weighted percentiles from GHFHs from all BTPs is called PPIF ensemble

hereinafter.

Estimates from temperature profiles and from heat flux profiles using the PPI and Standard approaches need to include inversions from the same number of BTPs to obtain the same geographical representation of surface temperature and heat flux changes. This requirement reduces the number of borehole considered in the analysis to 1060, 1072 and 1074 for the 25 yr, 40 yr and 50 yr inversion models, respectively, since not all BTPs provide with GSTHs and GHFHs complying with all criteria

explained in Section 3.3.2.

### 3.4  Flux Estimates from Surface Temperatures

The relationship between surface flux (q) and a temporal series of surface temperatures can be expressed as (Wang and Bras, 1999; Beltrami, 2001)

$$q_{t_N} = \frac{2\lambda}{\sqrt{\pi \kappa \Delta t}} \sum_{i=1}^{N-1} \left\{ (T_i - T_{i+1}) \left( \sqrt{N-i} - \sqrt{N-i-1} \right) \right\}, \tag{19}$$

where $\Delta t$ is the length of the time steps and $T_i$ is surface temperature at the $i$th time step. We estimate ground heat flux histories at the surface from GSTHs retrieved from both the Standard and PPI approaches. Thermal properties for estimating heat fluxes from GSTHs obtained with the Standard inversion approach are set to $\lambda = 3 \, \mathrm{W \, m^{-1} K^{-1}}$ and $\kappa = 1 \times 10^{-6} \, \mathrm{m^2 \, s^{-1}}$, while thermal properties for estimating heat fluxes from GSTHs included in the PPIT ensemble are set as those associated to the corresponding individual GSTH. Heat flux estimates are also provided using Equation 19 and CRU temperature data in order

to create the histogram of heat flux changes displayed in Figure S1, considering the same thermal properties as in heat flux estimates from GSTHs retrieved by the Standard approach.

## 4  Results

Ground surface temperature histories estimated using a 25 yr inversion model together with the Standard approach and the new PPIT ensemble show temperature increases that are particularly large during the second half of the 20[th] century in comparison

with preindustrial conditions (Figure 3a). This is in agreement with meteorological observations of surface air temperatures (red and orange lines in the mentioned figure), as well as with previous studies using both borehole temperature profiles and proxy data (Pollack et al., 1998; Huang et al., 2000; Beltrami, 2002a; Pollack and Smerdon, 2004; Fernández-Donado et al., 2013; Masson-Delmotte et al., 2013). Both approaches used to retrieve GSTHs from temperature profiles display a remarkable agreement during the whole period, as well as similar temperature changes to those shown by CRU surface temperatures for the

observational period. Global mean temperature changes between 1950-1975 CE and 1975-2000 CE reach 0.3 K for the PPIT





ensemble and $0.4$ K for the Standard approach (Table 1), with mean temperature changes from CRU data yielding approximately $0.4$ K using both the entire dataset and locations and dates containing BTP inversions. GSTHs present slightly higher temperature changes since preindustrial times than previously reported, with results ranging from $1.0 \pm 0.1$ K to $1.2 \pm 0.2$ K for the last part of the $20^{th}$ century considering results from the three inversion models (Tables 1, S1 and S2) in comparison

to the $\sim 0.9$ K reported in previous works (Huang et al., 2000; Harris and Chapman, 2001; Beltrami, 2002a; Pollack and Smerdon, 2004).

As in the case of surface temperature histories, the three approaches providing ground heat flux histories from BTP measurements are in good agreement during the entire period, although presenting higher uncertainties than for temperatures (Figure 3b and Table 1). Global results from Beltrami et al. (2002) are also displayed in Figure 3b (purple line), achieving similar

values in comparison with GHFHs derived by the Standard, PPIT and PPIF approaches except for the second half of the $20^{th}$ century. Global heat flux change achieves $70 \pm 20$ mW m$^{-2}$, $60 \pm 50$ mW m$^{-2}$ and $60 \pm 40$ mW m$^{-2}$ for the Standard, PPIT and PPIF ensembles, respectively (Table 1), in contrast to the $39 \pm 4$ mW m$^{-2}$ presented in Beltrami et al. (2002) and the $\sim 33$ mW m$^{-2}$ from Beltrami (2002a). The large number of recently acquired profiles included in our analysis may explain the larger flux estimates in comparison with previous works, since BTP measurements recorded before the 1980s did not capture

the large disturbances in the surface energy budget from recent decades (Stevens et al., 2008). Global changes in GHC were estimated for the Standard, PPIT and PPIF heat flux histories by scaling these fluxes to the continental areas except Antarctica and Greenland, where there are no BTP measurements. GHC changes of $15 \pm 5$ ZJ, $10 \pm 10$ ZJ and $13 \pm 8$ ZJ (1 ZJ $= 10^{21}$ J) are obtained for the period 1950-2000 CE using the Standard, PPIT and PPIF approaches, respectively, in comparison with the $9 \pm 1$ ZJ in Beltrami et al. (2002) and the 7 ZJ in Beltrami (2002a). As expected, these estimates of continental heat storage

are larger than previously reported since the heat flux histories also present higher values. The small uncertainty for heat flux histories, and therefore for estimates of continental heat storage, showed by the Standard and PPIT ensembles at the beginning of the period (Figure 3) is artificially imposed by Equation 19, since the heat flux estimate for the first temporal step is set to zero by default. Therefore, the PPIF ensemble is providing a more realistic estimate of the uncertainty in the global GHFHs and GHC estimates for the first half of the period, with larger uncertainties for the three approaches in the second half of the

period.

Although the borehole database used here contains BTP measurements recorded after 2000 CE, results are shown until the end of the $20^{th}$ century, since the number of available logs decreases sharply afterwards and the remaining profiles are located mainly at high latitudes in North America and Australia (Figure 1). We use the trend for the period 1970-2000 CE to extrapolate the heat flux histories until 2018 CE, providing with an estimate of the accumulated heat content in the continental

subsurface from 1960 CE to the present (Figure 4). The global mean change of heat flux for the entire period is approximately $90$ mW m$^{-2}$ considering all inversion approaches, while the global heat flux change since 2000 CE is $\sim 120$ mW m$^{-2}$. Thus, the accumulated heat within the global continental subsurface obtained from these flux estimates achieve 20 ZJ for the entire period and $\sim 9$ ZJ for 2000-2018 CE. That is, if the global heat flux increase during the first decades of the $21^{st}$ century resembled the trend of the period 1970-2000 CE, half of the total increase in energy storage within the continental subsurface





in the last fifty-eight years would have occurred during the last two decades, a remarkably similar result in comparison with the accelerated ocean heat uptake in the last decades (Gleckler et al., 2016; Cheng et al., 2017, 2019).

## 5   Discussion

Ground surface temperature and ground heat flux histories retrieved by the three inversion models used here achieve similar evolutions since preindustrial times, and yield similar estimates of ground heat content for all continental areas without con-
sidering Antarctica and Greenland (Figures 3, S2 and S3, and Tables 1, S1 and S2). Nonetheless, the surface temperature, heat flux and heat storage results are larger than previous global estimates of GSTHs, GHFHs and GHC from borehole data (Pollack et al., 1998; Huang et al., 2000; Beltrami, 2002a; Beltrami et al., 2002; Pollack and Smerdon, 2004). The main reason for the higher values reported here is the inclusion of additional temperature profiles measured at more recent dates than those employed in the literature, since logs acquired after the 1980s and 1990s recorded larger changes in the subsurface thermal
regime due to larger variations in the surface energy balance (Stevens et al., 2008). That is, more than 250 high-quality logs have been measured or made available for the community since the early 2000s, including profiles from scarcely represented areas in the southern hemisphere. Additionally, there have been improvements in the aggregation and treatment of borehole profiles contributing to the differences between our estimates and previous works (Beltrami et al., 2015b). We have truncated all logs to the same depth before performing the analysis in contrast to previous studies, which used profiles including a range
of bottom depths, therefore including GSTH and GHFH estimates with different periods of reference.

The larger differences in uncertainties in heat flux estimates from the PPIT ensemble in comparison with those from the PPIF ensemble are caused by the criterium to discard unrealistic inversions in the PPI approach (Figures 3b, S2b and S3b). That is, the heat flux estimates for the PPIT ensemble were not filtered out using the flux criterium ($0.51 \ \mathrm{W\,m^{-2}}$) of the PPI approach but the temperature criterium ($2.57 \ \mathrm{K}$). Applying these different criteria is necessary since heat flux estimates from
the PPIT ensemble result from applying Equation 19 to the previously retrieved surface temperature histories, while the heat flux histories considered in the PPIF ensemble result from direct inversions of heat flux profiles, as explained in Section 3.3.2.

Borehole temperature profiles present a unique ability to integrate multi-centennial changes in the surface energy balance (Beltrami, 2002b), which makes borehole inversions an important source of information about preindustrial conditions. The depth range considered here (from $15 \ \mathrm{m}$ to $300 \ \mathrm{m}$) allows to retrieve information from $\sim 700$ years before the logging date
of each log, i.e., several centuries before the industrialization. Thus, all surface temperature histories displayed in Figures 3a, S2a and S3a are relative to approximately 1300-1700 CE, as the subsurface quasi-equilibrium profile is estimated here from the $200$-$300 \ \mathrm{m}$ depth range for all profiles (Cuesta-Valero et al., 2019). The ground surface temperature increases relative to preindustrial conditions from the three PPIT ensembles analyzed here are $\sim 1.0 \ \mathrm{K}$ for the last part of the $20^{\mathrm{th}}$ century (Tables 1, S1 and S2). This is not, however, an estimate of the global temperature change, since land temperature changes at a
higher pace than the temperature at the surface of the ocean due to their different thermal properties. The ratio between land temperature change and ocean temperature change is estimated in Harrison et al. (2015) based on an ensemble of long-term CGCM simulations performed under different external forcings, resulting in land temperature changes $\sim 2.36$ times larger





than ocean temperature changes. Thus, the corresponding ocean temperature change to the land temperature change retrieved from borehole temperature profiles can be approximated as $\sim 0.4$ K, which yields an approximated global surface temperature
change of $\sim 0.7$ K since preindustrial times. Such a temperature change from preindustrial conditions is in good agreement with the estimates of $0.55 - 0.8$ K discussed in Hawkins et al. (2017) using observations, CGCM simulations and proxy databases, even for a preindustrial period much further in the past in comparison with the periods analyzed in Schurer et al. (2017).

These new estimates of continental heat storage and ground heat flux from BTP inversions have implications for the assessment of the Earth's energy budget and for the comparison with CGCM simulations. Although the ocean is still the largest
component of the Earth's energy budget, the contribution of the continental subsurface is higher than previously reported, reinforcing the necessity of monitoring and accounting for the rest of components. Furthermore, previous assessments have shown that CGCM simulations are unable to represent changes in continental heat storage due to their shallow land surface model components (Stevens et al., 2007; MacDougall et al., 2008; Cuesta-Valero et al., 2016). The new GHC estimates emphasize the demand for deeper subsurfaces in CGCMs in order to generate global transient simulations capable of correctly reproducing
the Earth's energy budget.

The distribution of BTP measurements used in this analysis is specially scarce in zones of Africa, South America and the Middle East, which may rise doubts about the global representativity of the assembled borehole dataset. Previous works have assessed the spatial distribution of BTP measurements using transient climate simulations performed by CGCMs at millennial time scales (González-Rouco et al., 2006; González-Rouco et al., 2009; García-García et al., 2016; Melo-Aguilar et al., 2019),
and borehole databases aggregated using different techniques (Beltrami and Bourlon, 2004; Pollack and Smerdon, 2004), with all studies concluding that the effects of limited regional sampling on estimates of global changes should be minor. Additionally, surface air temperatures from CRU data present markedly similar values considering both the full domain, and locations and dates containing BTP inversions (see red and orange lines in Figure 3), supporting the claim that borehole temporal and spatial distributions are representative of global conditions. Nevertheless, repeating measurements at borehole sites previously logged
as well as obtaining new records at zones with reduced density of BTP data would improve the global estimates of ground surface temperature and ground heat flux histories from borehole temperature profiles.

## 6 Conclusions

The magnitude of the retrieved changes in ground surface temperature in this analysis supports the claim that the Earth's surface has warmed by $\sim 0.7$ K since preindustrial times. The new estimates also reveal that the continental subsurface has
stored more energy during the last part of the 20[th] century than previously reported, reaching around 12 ZJ. This evidences the need for including deeper land surface model components in CGCM transient simulations in order to correctly reproduce the land component of the Earth's energy budget, as well as potentially powerful carbon feedbacks related to energy-dependent processes of the continental subsurface, such as the stability of the soil carbon pool and permafrost evolution.



*Data availability.* Data from the Climatic Research Unit (CRU) of East Anglia University can be accessed at http://doi.org/10/gcmcz3. Bore-
hole data can be downloaded from: NOAA server for a global dataset (ftp://ftp.ncdc.noaa.gov/pub/data/paleo/borehole/), Jaume-Santero et al.
(2016) for North America (doi:10.6084/m9.figshare.2062140), Suman et al. (2017) for Tasmania (https://doi.org/10.5194/cp-13-559-2017-supplement),
and Pickler et al. (2018) for Chile (doi:10.6084/m9.figshare.5220964.v2).

*Author contributions.* FJCV analyzed the borehole data, developed the PPI technique applied to characterize uncertainties in borehole inver-
sions, and produced all results and figures. All authors contributed to the interpretation and discussion of results. FJCV wrote the manuscript
with continuous feedback from all authors.

*Competing interests.* The authors declare that they have no conflict of interest.

*Acknowledgements.* This work was supported by grants from the Natural Sciences and Engineering Research Council of Canada Discovery
Grant (NSERC DG 140576948), the Canada Research Chairs Program (CRC 230687), and the Canada Foundation for Innovation (CFI)
to H. Beltrami. H. Beltrami holds Canada Research Chair in Climate Dynamics. A.G.G. and F.J.C.V. are funded by H. Beltrami's Canada
Research Chair program, the School of Graduate Students at Memorial University of Newfoundland and the Research Office at St. Francis
Xavier University.



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



**Table 1.** Global mean estimates of ground surface temperature, ground heat flux at the surface and ground heat content within continental subsurface from borehole temperature profiles. Values display the mean and 95% confidence interval for each time period from estimates using the Standard inversion approach (Standard), the new PPI approach (PPIT) and the new PPI approach applied to subsurface flux profiles (PPIF). All the inversions were performed using a model of 25 years per time step. Temperatures in K, fluxes in $\mathrm{mW\,m^{-2}}$ and heat content in ZJ.

| | Temperatures | | Heat Fluxes | | | Heat Storage | | |
|---|---|---|---|---|---|---|---|---|
| Period (CE) | Standard | PPIT | Standard | PPIT | PPIF | Standard | PPI | PPIF |
| 1975-2000 | $1.2 \pm 0.2$ | $1.1 \pm 0.3$ | $100 \pm 20$ | $70 \pm 70$ | $80 \pm 40$ | $10 \pm 2$ | $8 \pm 8$ | $9 \pm 5$ |
| 1950-1975 | $0.8 \pm 0.2$ | $0.8 \pm 0.5$ | $40 \pm 40$ | $40 \pm 80$ | $40 \pm 60$ | $4 \pm 4$ | $4 \pm 8$ | $4 \pm 6$ |
| 1925-1950 | $0.6 \pm 0.2$ | $0.6 \pm 0.6$ | $30 \pm 30$ | $40 \pm 60$ | $30 \pm 60$ | $3 \pm 4$ | $4 \pm 7$ | $3 \pm 6$ |
| 1900-1925 | $0.5 \pm 0.3$ | $0.4 \pm 0.5$ | $30 \pm 20$ | $30 \pm 40$ | $20 \pm 50$ | $3 \pm 2$ | $3 \pm 4$ | $2 \pm 5$ |
| 1875-1900 | $0.4 \pm 0.3$ | $0.3 \pm 0.5$ | $30 \pm 20$ | $20 \pm 40$ | $20 \pm 40$ | $3 \pm 2$ | $2 \pm 4$ | $2 \pm 5$ |
| 1850-1875 | $0.3 \pm 0.3$ | $0.2 \pm 0.5$ | $20 \pm 20$ | $10 \pm 30$ | $10 \pm 40$ | $2 \pm 2$ | $1 \pm 3$ | $1 \pm 4$ |
| 1825-1850 | $0.2 \pm 0.3$ | $0.1 \pm 0.4$ | $20 \pm 20$ | $9 \pm 30$ | $10 \pm 30$ | $2 \pm 2$ | $0.9 \pm 3$ | $1 \pm 3$ |
| 1800-1825 | $0.1 \pm 0.2$ | $0.07 \pm 0.3$ | $10 \pm 20$ | $6 \pm 20$ | $7 \pm 20$ | $1 \pm 2$ | $0.7 \pm 3$ | $0.8 \pm 2$ |
| 1775-1800 | $0.04 \pm 0.2$ | $0.03 \pm 0.3$ | $10 \pm 20$ | $5 \pm 20$ | $5 \pm 20$ | $1 \pm 2$ | $0.5 \pm 2$ | $0.5 \pm 2$ |
| 1750-1775 | $-0.007 \pm 0.1$ | $0.002 \pm 0.3$ | $7 \pm 10$ | $3 \pm 20$ | $3 \pm 20$ | $0.7 \pm 1$ | $0.4 \pm 2$ | $0.3 \pm 2$ |
| 1725-1750 | $-0.04 \pm 0.07$ | $-0.02 \pm 0.3$ | $5 \pm 10$ | $3 \pm 10$ | $2 \pm 20$ | $0.5 \pm 1$ | $0.3 \pm 1$ | $0.2 \pm 3$ |
| 1700-1725 | $-0.07 \pm 0.04$ | $-0.03 \pm 0.3$ | $3 \pm 9$ | $2 \pm 10$ | $1 \pm 30$ | $0.3 \pm 0.9$ | $0.2 \pm 1$ | $0.1 \pm 3$ |
| 1675-1700 | $-0.08 \pm 0.06$ | $-0.04 \pm 0.3$ | $2 \pm 7$ | $1 \pm 8$ | $0.7 \pm 30$ | $0.2 \pm 0.7$ | $0.1 \pm 0.8$ | $0.07 \pm 3$ |
| 1650-1675 | $-0.10 \pm 0.08$ | $-0.04 \pm 0.3$ | $1 \pm 5$ | $0.9 \pm 6$ | $0.3 \pm 30$ | $0.1 \pm 0.5$ | $0.09 \pm 0.6$ | $0.03 \pm 3$ |
| 1625-1650 | $-0.1 \pm 0.1$ | $-0.05 \pm 0.3$ | $0.6 \pm 4$ | $0.5 \pm 4$ | $0.07 \pm 30$ | $0.07 \pm 0.4$ | $0.05 \pm 0.4$ | $0.007 \pm 4$ |
| 1600-1625 | $-0.1 \pm 0.1$ | $-0.05 \pm 0.3$ | $0.08 \pm 2$ | $0.2 \pm 2$ | $-0.2 \pm 40$ | $0.009 \pm 0.2$ | $0.02 \pm 0.2$ | $-0.02 \pm 4$ |



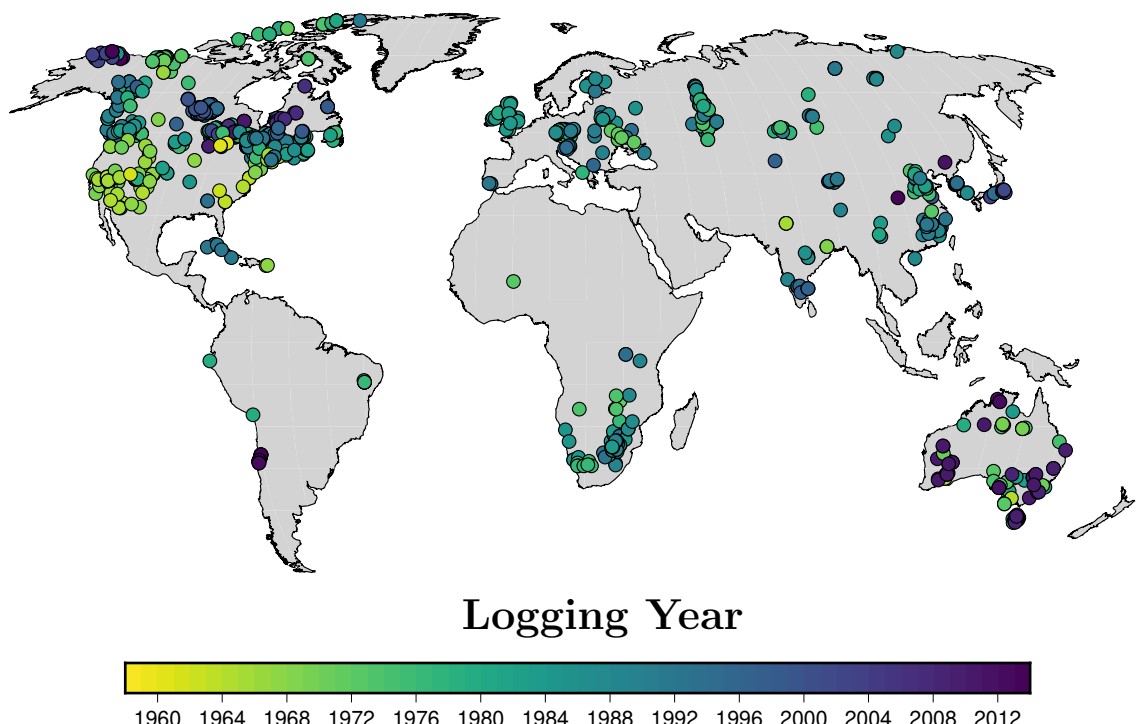

**Figure 1.** Logging years of the 1079 boreholes considered in the analysis.

**Figure 2.** Borehole temperature profile measurements at Fox Mine (CA_9519), Manitoba (Canada) as an example to explain the inversion approaches in this study. (a) Observed original profile (black dots) as well as the estimated subsurface quasi-equilibrium temperature profile (black line) and the two extremal temperature profiles (red and blue lines) displaying the 95% uncertainty in determining the quasi-equilibrium profile. All three equilibrium profiles were estimated from the linear regression analysis of the deepest part of the measured profile (from 200 m to 300 m, grey zone). (b) Anomaly profiles estimated by subtracting the three equilibrium profiles to the original temperature profile. (c) As in (b), but including the 243 synthetic profiles generated from the corresponding ground surface temperature histories constituting the PPI ensemble of this borehole (red, blue and black shades). (d) Final ensemble of ground surface temperature histories considered for estimating the 5th, 50th and 95th weighted percentiles for this borehole. Each history is weighted depending on its performance against the corresponding anomaly profile (pannel c).





**Figure 3.** Global ground surface temperature histories (a) and global ground heat flux histories at the surface (b) from borehole temperature profiles using the Standard approach (black), the new PPI approach (PPIT, blue) and the new PPI approach applied to the corresponding heat flux profiles (PPIF, light blue). All inversions were performed using a 25 yr inversion model. (c) Percentage of total borehole inversions with time. Surface air temperature anomalies relative to 1961-1990 CE from CRU data are also displayed, including results from the entire database (red) and results from locations and dates containing borehole inversions (orange). The CRU series have been adjusted to have the same mean than the results from the Standard approach for the period 1950-1970 CE.





**Figure 4.** Global ground heat flux histories (a) and ground heat content accumulated since 1960 CE (b) from borehole temperature profiles using the Standard approach (black), the new PPI approach (PPIT, blue) and the new PPI approach applied to the corresponding heat flux profiles (PPIF, light blue). All inversions were performed using a 25 yr inversion model. Data since 2001 CE to 2018 CE are extrapolated using the trend for the period 1971-2000 CE.