# Peer review of "Long-Term Global Ground Heat Flux and Continental Heat Storage from Geothermal Data"

_Climate of the Past, 2020_

## Referee Comment (RC1) · Anonymous Referee #1 · 15 Jul 2020

Summary:

The manuscript presents an estimation of past ground heat flux and past surface temperatures over the last few centuries based on measured borehole temperature profiles. The main objective of the analysis is to estimate the history of vertical heat flux into the ground, in the more general framework of the global energy fluxes perturbed by anthropogenic climate change. The methodology of deriving past surface temperatures from borehole temperature profiles is well established. The novelty in this study is threefold: the shifted focus towards the surface heat fluxes, the estimation of uncertainties, and the expansion of the available data base. The main conclusion is that the

ground heat flux estimate from borehole profiles has larger than had previously estimated. The authors claim that this component of the energy fluxes is important within the climate system.

Recommendation:

Some revisions necessary, but I think this is a valuable contribution to Climate of the Past. The manuscript is generally well written - although some sections would benefit from a revision.

General comments:

1) I found Section 2 too detailed. It will certainly help readers with a more superficial background on borehole climatology, but I think that this section can be compressed, displaying the main ideas and the important technical details that are used later on in the manuscript. For instance, I do not think it is necessary to display equation 11 in such level of detail. A matrix equation should suffice

2) In contrast, section 3 should include the new methodological aspects of the direct heat flux inversion. Here, either I missed something or something is indeed missing. On the one hand, the manuscript alludes to a direct inversion of the flux profiles (equation 18) to heat flux histories, using also the Perturbed Parameter approach (line 266) But the methodology for the direct inversion of heat flux histories is not explained, at least I could not find it in the manuscript. The PPI approach has been explained for the temperature inversions, not the heat flux inversions. Perhaps, it is so obvious that it does not need an explanation, but to me it is not that clear. In case I misunderstood something here, it is likely than an average reader will also get confused. There is an imbalance between the level of detail presented for the temperature history inversions and for the heat flux inversions.

On the other hand, the manuscript also used ground heat flux histories derived from the inversion of ground temperature histories, equation (19). There are then apparently

two reconstructions of the ground heat flux histories, one by a 'direct inversion method' and one based on the reconstructed surface temperatures. And yet a third estimation for the recent period using the CRU temperatures. If this is true, it should be clearly indicated. Please consider labeling these three products to guide the reader.

3) The approach leading to the weighting scheme in equation 17 can be problematic. I am not saying it is wrong, but a more versed statistician than me may complain. In essence, what the authors are doing is applying Bayesian scheme to estimate the inversion uncertainties. They assume a prior distribution of some model parameters, which are then passed through the model to produce temperature profiles, and these synthetic profiles are weighted by the likelihood (17). The problem is that there are hidden assumptions in this approach that are not explicitly stated. Are the initial model parameters a priori equally probable ? Without that assumption it is not possible to attach posterior probabilities to the synthetic profiles and to the model parameters. A more sophisticated, fully Bayesian approach could include a Monte Carlo Markov chain sampling of those posterior probabilities and of the temperature histories , in which their values are varied in a more systematic scheme. In any case, the hidden assumptions that authors are making about the relative probabilities of the assumed model parameters need to explicitly stated .

A second comment is that I guess that sigma in equation 17 is also depth-dependent . If not, please state clearly. If yes. would it have an impact ?

4) The main claim of the study is that the ground heat flux cannot be neglected. I miss a more direct comparison with the ocean heat flux, so that the reader gets a clearer idea. Probably, the ocean heat flux is much larger but the authors can more clearly elaborate their point.

Particular comments

5) line 30 'and sea level rise'

This is the major consequence of increase in ocean heat storage, so it is surprising that it is included with 'The rest of the components in the climate system'

6) line 63 'the model resolution for obtaining stable solutions' .

the vertical resolution

7) line 75' These results also support previous estimates of temperature change since preindustrial times based on meteorological observations and CGCM simulations, using estimates from an independent source of data and considering the most distant period of time to determine preindustrial conditions to our knowledge.

This paragraph is unclear and hard to read

8) line 82 In borehole climatology, the continental subsurface is typically represented as a semi-infinite homogeneous half-space without internal sources of heat, where energy exchanges at the land surface and heat flux from the Earth's interior are considered as the'

half-space is not a well-defined term. Please, rephrase this paragraph more clearly

9) line 212 'the 95% confidence interval (two standard deviations) of the anomaly profile'

This is only (approximately ) true if the distribution is gaussian.

---

## Referee Comment (RC2) · Anonymous Referee #2 · 26 Sep 2020

Review of Cuerta-Valero et al Long-Term Global Ground Heat Flux and Continental Heat Storage from Geothermal Data

This paper represents a useful update and expansion of a large body of work that uses borehole temperature measurements to estimate surface temperature changes and accumulation of heat in the upper few hundred metres of the Earth's crust, both associated with recent climate change.

Advances in the paper include (a) the addition of additional borehole temperature data, and (b) a new approach to the inversion of the borehole data that produces better estimates of the uncertainties.

The introduction section is a particularly useful, comprehensive summary of work in this area with an extensive reference list.

Figure 3a, the updated ground temperature history from 1580 CE to present with uncertainty estimates is very important. It is shown in comparison to previous ground temperature histories and the meteorological record back to ~1900 and should be widely used as a constraint in climate reconstructions. The authors perhaps should make a stronger point that Fig3a (and the analysis that results in Fig 3a) shows about 0.4K of warming from pre-industrial times to the start of the observational meteorological record around 1880. And the total land surface warming to present time (Fig 3a) is close to 1.4K.

In view of that number I don't understand the sentence in the conclusions that reads "The magnitude of the retrieved changes in ground surface temperature in this analysis supports the claim that the Earth's surface has warmed by âĹij 0.7 K since preindustrial times." Nor the sentence in the abstract that includes "land temperature changes of 1 K ... during the last part of the 20th century relative to preindustrial times." These statements should be consistent with each other and with Fig. 3a.

Attention to the following details would improve the manuscript. 1. In Eqn 1, R is not a thermal depth which would have dimensions of length. It is in fact the thermal resistance with units $m^2$ K /W. 2. In section 3.1, the criteria for accepting a borehole temperature log of 1 measurement in the 15-100m depth range and at least 3 measurements in the 250-310 m depth range seems pretty loose. It would be good to know why such a fairly lax criteria was chosen and how many sites creep into the data set as a result. 3. It is a personal style, but I would prefer fewer acronyms. Are the following all necessary: GHC, BTP, GSTH, GHFH, PPI, RMSE, PPIT?

Overall this paper is a very useful contribution to the climate change literature.

---

## Editor Comment (EC1) · Nerilie Abram (Editor) · 1 Oct 2020

Dear Francisco Jose Cuesta-Valero and co-authors,

Your manuscript has now received two reviewer reports and the interactive discussion phase has ended. The reviewers both find that your work is a useful contribution to the climate science literature and they provide suggestions for revising the manuscript.

Please now proceed with providing responses to the reviewer comments in the interactive discussion to continue the peer-review evaluation process.

Sincerely, Nerilie Abram

---

## Author Comment (AC1) · 2 Oct 2020

Response to Reviewers Document for "Long-Term Global Ground Heat Flux and Continental Heat Storage from Geothermal Data" by Francisco José Cuesta-Valero, Almudena García-García, Hugo Beltrami, Fidel González-Rouco and Elena García-Bustamante.

We thank the Reviewers for their thoughtful and constructive feedback.

This Response to Reviewers file provides a complete documentation of the changes made in response to each individual Reviewer's comment. Reviewers'

[Figure]

comments are shown in plain text. Author responses are shown in bold blue text. Corrections within the revised manuscript are shown in blue text. All line numbers in this file refer to locations in the revised manuscript with changes marked unless indicated otherwise.

Reviewer 1

Summary:

The manuscript presents an estimation of past ground heat flux and past surface temperatures over the last few centuries based on measured borehole temperature profiles. The main objective of the analysis is to estimate the history of vertical heat flux into the ground, in the more general framework of the global energy fluxes perturbed by anthropogenic climate change. The methodology of deriving past surface temperatures from borehole temperature profiles is well established. The novelty in this study is threefold: the shifted focus towards the surface heat fluxes, the estimation of uncertainties, and the expansion of the available data base. The main conclusion is that the ground heat flux estimate from borehole profiles has larger than had previously estimated. The authors claim that this component of the energy fluxes is important within the climate system.

Recommendation:

Some revisions necessary, but I think this is a valuable contribution to Climate of the Past. The manuscript is generally well written - although some sections would benefit from a revision.

General comments:

1) I found Section 2 too detailed. It will certainly help readers with a more superficial background on borehole climatology, but I think that this section can be compressed, displaying the main ideas and the important technical details that are used later on in the manuscript. For instance, I do not think it is necessary to display equation 11 in

such level of detail. A matrix equation should suffice.

**Indeed, Section 2 consists of a detailed description of borehole methodology. We have reduced the level of detail in Section 2.3 as suggested by the reviewer, although the rest of Section 2 only contains minor adjustments, since we described the most important concepts of borehole climatology to improve the overall clarity and reproducibility of the work described in the article.**

2) In contrast, section 3 should include the new methodological aspects of the direct heat flux inversion. Here, either I missed something or something is indeed missing. On the one hand, the manuscript alludes to a direct inversion of the flux profiles (equation 18) to heat flux histories, using also the Perturbed Parameter approach (line 266) But the methodology for the direct inversion of heat flux histories is not explained, at least I could not find it in the manuscript. The PPI approach has been explained for the temperature inversions, not the heat flux inversions. Perhaps, it is so obvious that it does not need an explanation, but to me it is not that clear. In case I misunderstood something here, it is likely than an average reader will also get confused. There is an imbalance between the level of detail presented for the temperature history inversions and for the heat flux inversions.

**The inversion method used to retrieve ground heat flux histories from heat flux profiles is the same as the one for estimating ground surface temperature histories from temperature profiles. Hence, we described both inversion procedures in the same section (Section 3.3.2). Nevertheless, we agree with the reviewer about the confusion that this may cause on the reader. Thus, we have rearranged the text into two specific sections on the new version of the manuscript, one for describing the inversion of temperature profiles, and another one for describing the inversion of heat flux profiles (Sections 3.3.2 and 3.3.3, respectively).**

On the other hand, the manuscript also used ground heat flux histories derived from the inversion of ground temperature histories, equation (19). There are then apparently

two reconstructions of the ground heat flux histories, one by a 'direct inversion method' and one based on the reconstructed surface temperatures. And yet a third estimation for the recent period using the CRU temperatures. If this is true, it should be clearly indicated. Please consider labeling these three products to guide the reader.

**We have changed the confusing terms on the text, figures and tables, incorporating the variable and the method used to obtain the variable in the name of each estimate. Thus, the temperature and flux data from the CRU product are now named SAT_CRU and GHF_CRU, respectively; the temperature, flux and heat estimates from borehole temperature profiles using the Standard method are named GST_Standard, GHF_Standard and GHC_Standard, respectively; the temperature, flux and heat estimates from borehole temperature profiles using the PPI method are named GST_PPIT, GHF_PPIT and GHC_PPIT, respectively; and the flux and heat estimates from borehole flux profiles using the PPI method are named GHF_PPIF and GHC_PPIF, respectively. We have also included a new appendix including the definition of all these acronyms (Appendix A, page 17).**

3) The approach leading to the weighting scheme in equation 17 can be problematic. I am not saying it is wrong, but a more versed statistician than me may complain. In essence, what the authors are doing is applying Bayesian scheme to estimate the inversion uncertainties. They assume a prior distribution of some model parameters, which are then passed through the model to produce temperature profiles, and these synthetic profiles are weighted by the likelihood (17). The problem is that there are hidden assumptions in this approach that are not explicitly stated. Are the initial model parameters a priori equally probable? Without that assumption it is not possible to attach posterior probabilities to the synthetic profiles and to the model parameters. A more sophisticated, fully Bayesian approach could include a Monte Carlo Markov chain sampling of those posterior probabilities and of the temperature histories, in which their values are varied in a more systematic scheme. In any case, the hidden assumptions that authors are making about the relative probabilities of the assumed model param-

eters need to explicitly stated.

**We tried to develop a method allowing us to include more uncertainty terms in the analysis that previous studies using the inversion methodology described in Sections 2. The measurement error and the uncertainty in the determination of the equilibrium temperature profile have been included in previous studies (e.g., Beltrami et al., 2015a; Jaume-Santero et al., 2016; Pickler et al., 2016, 2018), but other sources of uncertainty remained unaddressed. The objective of the PPI method is to comprehensibly estimate the uncertainty due to as many factors affecting the inversion of the profiles as possible. To this end, we explore the range of reasonable thermal properties within borehole temperature profiles, as the thermal properties are typically unknown at most borehole locations. We also attempt to include the uncertainty related to the number of eigenvalues employed in the inversion, as there is no general rule to determine the total number of eigenvalues that should be conserved in the process.**

**Therefore, we are not trying to perform a bayesian inversion, we just generalize a typical inversion method to account for more sources of uncertainty than in previous studies using the evaluation of large ensembles of climate model simulations as inspiration (Knutti et al., 2017). Even more, our method is markedly different to a bayesian inversion, such as the one used in Shen and Beck, (1991) and Hopcroft et al., (2007). Nevertheless, it has been shown that all inversion methods retrieve similar surface signals from the same subsurface profiles (Shen et al., 1992).**

A second comment is that I guess that sigma in equation 17 is also depth-dependent. If not, please state clearly. If yes, would it have an impact?

**We indicated that sigma corresponds with the typical measurement error in borehole profiles, and therefore it is constant (line 248 of the original manuscript, line 259 on the new version of the manuscript). As stated in Knutti et al., (2017), the**

**sigma parameter in Eq. 16 (Eq. 17 in the original manuscript) is just a value determining what RSMEs are consider to be large or small. Therefore, the results will vary if changing sigma. Indeed, we could define a sigma that depends on depth, but we think that using a constant sigma improves the clarity of the method, as this is the first borehole study using it. A future study may evaluate the effect of this parameter on the retrieved inversions, but that is out of the scope of this work.**

4) The main claim of the study is that the ground heat flux cannot be neglected. I miss a more direct comparison with the ocean heat flux, so that the reader gets a clearer idea. Probably, the ocean heat flux is much larger but the authors can more clearly elaborate their point.

**We discussed the observed proportions of heat within the ocean and the continental subsurface in the Introduction of the original manuscript (lines 22-23 of the original manuscript). Nevertheless, we have included a comparison of the new estimates of ground heat flux at the surface presented in the manuscript with the ocean heat flux and the rest of terms of the Earth's heat inventory (lines 404-409).**

Particular comments

5) line 30 'and sea level rise'. This is the major consequence of increase in ocean heat storage, so it is surprising that it is included with 'The rest of the components in the climate system'.

**The reviewer is right, both ice melting and thermal expansion of the ocean contributes almost equally to sea level rise (Oppenheimer et al., 2019). We have changed the Introduction to clarify this point (lines 25-26 and 30-31).**

6) line 63 'the model resolution for obtaining stable solutions'. the vertical resolution.

**In fact, we were referring to the number of eigenvalues used to invert the bore-**

**hole profile. We have clarified this on the new version of the manuscript (line 62).**

7) line 75' These results also support previous estimates of temperature change since preindustrial times based on meteorological observations and CGCM simulations, using estimates from an independent source of data and considering the most distant period of time to determine preindustrial conditions to our knowledge. This paragraph is unclear and hard to read.

**We have changed the last paragraph of the Introduction on the new version of the text (lines 74-80).**

8) line 82 In borehole climatology, the continental subsurface is typically represented as a semi-infinite homogeneous half-space without internal sources of heat, where energy exchanges at the land surface and heat flux from the Earth's interior are considered as the'. Half-space is not a well-defined term. Please, rephrase this paragraph more clearly.

**By half-space we meant a mathematical space from the surface to an infinite depth, as the radius of the Earth is long enough to be considered infinite in this problem. We have included a more detailed definition on the new version of the text (lines 83-86).**

9) line 212 'the 95

**Indeed, we are considering gaussian distribution of errors in the measurements since we are using a linear regression analysis. We have reworded the terms on the text in order to avoid any confusion (lines 223-225).**

---

## Author Comment (AC2) · 2 Oct 2020

Response to Reviewers Document for "Long-Term Global Ground Heat Flux and Continental Heat Storage from Geothermal Data" by Francisco José Cuesta-Valero, Almudena García-García, Hugo Beltrami, Fidel González-Rouco and Elena García-Bustamante.

We thank the Reviewers for their thoughtful and constructive feedback.

This Response to Reviewers file provides a complete documentation of the changes made in response to each individual Reviewer's comment. Reviewers'

**comments are shown in plain text. Author responses are shown in bold blue text. Corrections within the revised manuscript are shown in blue text. All line numbers in this file refer to locations in the revised manuscript with changes marked unless indicated otherwise.**

Reviewer 2

Review of Cuerta-Valero et al Long-Term Global Ground Heat Flux and Continental Heat Storage from Geothermal Data.

This paper represents a useful update and expansion of a large body of work that uses borehole temperature measurements to estimate surface temperature changes and accumulation of heat in the upper few hundred metres of the Earth's crust, both associated with recent climate change. Advances in the paper include (a) the addition of additional borehole temperature data, and (b) a new approach to the inversion of the borehole data that produces better estimates of the uncertainties.

The introduction section is a particularly useful, comprehensive summary of work in this area with an extensive reference list. Figure 3a, the updated ground temperature history from 1580 CE to present with uncertainty estimates is very important. It is shown in comparison to previous ground temperature histories and the meteorological record back to 1900 and should be widely used as a constraint in climate reconstructions. The authors perhaps should make a stronger point that Fig3a (and the analysis that results in Fig 3a) shows about 0.4K of warming from pre-industrial times to the start of the observational meteorological record around 1880.

**The reviewers suggest an interesting result. We have included a comment on the Results section as indicated by the reviewer (lines 326-331).**

And the total land surface warming to present time (Fig 3a) is close to 1.4K. In view of that number I don't understand the sentence in the conclusions that reads "The magnitude of the retrieved changes in ground surface temperature in this analysis supports

the claim that the Earth's surface has warmed by 0.7 K since preindustrial times." Nor the sentence in the abstract that includes "land temperature changes of 1K... during the last part of the 20th century relative to preindustrial times." These statements should be consistent with each other and with Fig. 3a.

**Regarding the estimate of global temperature change since preindustrial times, that value is obtained as explained in the Discussion section (lines 357-372 of the original manuscript, lines 386-402 of the new version of the manuscript). The estimate is based on averages of land temperature reconstructions using the three inversion methods discussed in this analysis and a factor to convert land temperature changes into global temperature changes. That is, we use the averaged results for each inversion model as indicated in Tables 1, S1 and S2 to estimate the change in land temperature relative to preindustrial conditions –in this case, the mean temperature between 1300 CE and 1700 CE (lines 360-363 of the original manuscript, lines 389-391 of the new version of the manuscript). Then, we calculate global (land and ocean) temperature change by scaling the change in land temperatures to account for the probable change in ocean temperatures, resulting in an increase in global temperature of around 0.7 K since preindustrial times (lines 365-372 of the original manuscript, lines 397-399 of the new version of the manuscript).**

**We have added some changes in the Discussion to improve the clarity of the text (lines 386-402).**

Attention to the following details would improve the manuscript. 1. In Eqn 1, R is nota thermal depth which would have dimensions of length. It is in fact the thermal resistance with units mËĘ2 K /W.

**The reviewer is right, we have changed this on the new version of the text (line 94).**

2. In section 3.1, the criteria for accepting a borehole temperature log of 1 measurement in the 15-100m depth range and at least 3 measurements in the 250-310 m depth range seems pretty loose. It would be good to know why such a fairly lax criteria was chosen and how many sites creep into the data set as a result.

**Indeed, we included three criteria to select suitable borehole logs for our analysis: at least one temperature measurements between 15 m and 100 m to ensure the borehole profile recorded climate information near the logging year, at least one temperature measurement between 250 m and 310 m to ensure all temperature anomaly profiles include information about seven centuries before the logging date, and at least three temperature measurements between 200m and 300 m in order to perform a linear regression analysis.**

**We have changed the text in order to clearly explain why these criteria are applied and the number of logs excluded due to this filtering (lines 171 and 178-185).**

3. It is a personal style, but I would prefer fewer acronyms. Are the following all necessary: GHC, BTP, GSTH, GHFH, PPI, RMSE, PPIT?

**Indeed we used several acronyms to obtain a better flow in the original text. Furthermore, we have included additional acronyms in the new version of the manuscript to facilitate the interpretation of figures, responding the petition of the first reviewer. We have kept the acronyms that are more important to maintain the flow of the text, those necessary to understand the results, and those that are typical in scientific works, removing those that were superfluous. A new appendix (page 17) includes the remaining acronyms and their definition in order to improve the interpretation of results and the readability of the manuscript.**

Overall this paper is a very useful contribution to the climate change literature.

---

## Author Comment (AC3) · 2 Oct 2020

- Oppenheimer, M., B.C. Glavovic , J. Hinkel, R. van de Wal, A.K. Magnan, A. Abd-Elgawad, R. Cai, M. Cifuentes-Jara, R.M. DeConto, T. Ghosh, J. Hay, F. Isla, B. Marzeion, B. Meyssignac, and Z. Sebesvari (2019). Sea Level Rise and Implications for Low-Lying Islands, Coasts and Communities. In: IPCC Special Report on the Ocean and Cryosphere in a Changing Climate [H.-O. Pörtner, D.C. Roberts, V. Masson Delmotte, P. Zhai, M. Tignor, E. Poloczan-ska, K. Mintenbeck, A. Alegría, M. Nicolai, A. Okem, J. Petzold, B. Rama, N.M. Weyer (eds.)]. In press.

[Figure]

- **Shen, P. Y., and Beck, A. E. (1991). Least squares inversion of borehole temperature measurements in functional space, J. Geophys. Res., 96(B12), 19965-19979, doi:10.1029/91JB01883.**
* * *

---

## Author Comment (AC4) · 2 Oct 2020

**We thank the Reviewers for their thoughtful and constructive feedback.**

**We have addressed all reviewers' comments in this open discussion.**